# A Review of Periocular Botulinum Neurotoxin on the Tear Film Homeostasis and the Ocular Surface Change

**DOI:** 10.3390/toxins11020066

**Published:** 2019-01-24

**Authors:** Ren-Wen Ho, Po-Chiung Fang, Cheng-Hsien Chang, Yu-Peng Liu, Ming-Tse Kuo

**Affiliations:** 1Department of Ophthalmology, Kaohsiung Chang Gung Memorial Hospital and Chang Gung University College of Medicine, Kaohsiung 33302, Taiwan; wen6530@cgmh.org.tw (R.-W.H.); fangpc@cgmh.org.tw (P.-C.F.); 2Graduate Institute of Clinical Medicine, College of Medicine, Kaohsiung Medical University, Kaohsiung 80708, Taiwan; 3Department of Ophthalmology, China Medical University Hospital, China Medical University, Taichung 40402, Taiwan; ophchang@yahoo.com.tw

**Keywords:** Botulinum neurotoxin, dry eye, epiphora, inflammatory cytokine, lacrimal gland, lipid layer, ocular surface, tear

## Abstract

Clinical usage of botulinum neurotoxin (BoNT) in ophthalmology has dramatically increased since the 1980s and has become one of the most widely used agents for treating facial movement disorders, autonomic dysfunction and aesthetic wrinkles. Despite its high efficacy, there are some complications with periocular BoNT injections due to its chemodenervation effect. Among these, there is still controversy over the BoNT effect on tear film homeostasis and the ocular surface. A periocular BoNT injection could dry the eye by reducing tear production of the lacrimal gland and increase tear evaporation due to potential eyelid malposition and abnormal blinks. On the contrary, the injection of BoNT in the medial eyelids could treat dry eye disease by impairing lacrimal drainage. Regarding the ocular surface change, corneal astigmatism and high-order aberrations may decrease due to less eyelid tension. In conclusion, the entire awareness of the effect of BoNT and the patients’ ocular condition is crucial for successful and safe results.

## 1. Introduction

Clinical usage of botulinum neurotoxin (BoNT) in ophthalmology can trace back to the 1970s, which was first introduced by Dr Alan Scott as an alternative for strabismus treatment [1]. Thereafter, BoNT has been approved to be a safe and effective treatment for strabismus, blepharospasm and hemifacial spasm, autonomic dysfunction (e.g., gustatory lacrimation), and facial wrinkles [2,3,4,5]. 

BoNT is an exotoxin produced by the anaerobic, Gram-positive bacillus, *Clostridium botulinum*, and has seven subtypes, which are designated type A–G [6]. BoNT blocks the release of acetylcholine at cholinergic nerve terminals and the neuromuscular synapse, which causes temporary paralysis of target tissues [7]. Despite BoNT’s long-term efficacy and safety record for several ophthalmic diseases and aesthetic procedures, it has some potential complications, including blepharoptosis, brow ptosis, lagophthalmos, ectropion, and, less commonly, diplopia, caused by its chemodenervation effects on adjacent tissues. However, there is still controversy regarding the impact of BoNT on tear film changes, such as epiphora and dry eye. Several studies have found decreased lacrimal secretion after BoNT injection, whereas other studies have suggested that BoNT can improve dry eye symptoms and may be an alternative treatment for patients with dry eye. In this article, we review the published literature on the periocular use of BoNT and its effects on tear film homeostasis and the ocular surface.

## 2. Literature Search

We searched the PubMed database for articles published in English about the effects of BoNT on the tear film and ocular surface on 14th July 2018. The search strategy included the following keywords, “botulinum neurotoxin”, “botulinum toxin”, “tear”, “lipid layer thickness”, “lacrimal gland”, “meibomian gland”, “ocular surface”, and various combinations of these keywords. Animal studies in which BoNT was injected into the lacrimal glands were also included. Indications, BoNT dosages, injection sites, outcomes, and complications were also obtained.

## 3. Dry Eye Disease (DED)

According to the Tear Film and Ocular Surface Society (TFOS) Dry Eye Workshop (DEWS) II report [8], dry eye is a multifactorial disease of the ocular surface characterized by the loss of tear film homeostasis and accompanied by ocular symptoms in which tear film instability and hyperosmolarity, ocular surface inflammation and damage, and neurosensory abnormalities play etiological roles. Dry eye disease (DED) is usually classified as aqueous deficiency dry eye (ADDE) or evaporative dry eye (EDE) based on the predominant etiologies [8]. ADDE is a condition that affects the lacrimal glands, whereas EDE describes problems affecting the eyelids (e.g., meibomian gland dysfunction), ocular surface, or blink pattern, which result in rapid tear drying.

## 4. Results from Animal Studies of BoNT

Few studies have shown the inhibitory effects of BoNT on the lacrimal glands of mice and rabbits [9,10]. Moreover, BoNT has been used to induce dry eye animal models for a decade [11,12,13]. In 2006, a BoNT-induced mouse model of keratoconjunctivitis sicca was first established by Suwan-apichon et al. [9]. In this model, decreased aqueous tear production was observed with corneal fluorescein staining following the injection of 1.25 milliunits (mU) of BoNT-B into the lacrimal gland for one month. Demetriades et al. [10] analyzed the dose-dependent effects of BoNT in rabbits, which were divided into four groups receiving saline or 0.625, 1.25, or 2.5 U of BoNT-A. These authors found that 1.25 or 2.5 U of BoNT caused a significant reduction in aqueous tear production after one week. However, neither corneal staining nor changes in lacrimal gland architecture were found after BoNT injection. Both Park et al. [11] and Zhu et al. [12] injected 20 mU BoNT-B into the mouse lacrimal gland to induce dry eye and investigated lacrimal gland and ocular surface inflammatory cytokine expression. Zhu et al. found BoNT-B induced murine dry eye model revealed decreased tear production and elevated tumor necrosis factor (TNF)-α and interleukin (IL)-1β, which seemed to be the same as those in human with non-Sjogren’s syndrome keratoconjunctivitis sicca. Kim et al. [13] studied the tear volume, epidermal growth factor (EGF), and histology of the rabbit lacrimal gland after a 2.5 U BoNT-A injection. After BoNT-A injection, a significantly decreased tear volume (P = 0.012) and an increased EGF level (P = 0.011) and concentration (P = 0.012) were found without atrophic changes of the acini or fibrosis along the tubules. The elevated EGF level was considered to be a protective mechanism in response to corneal damage secondary to decreased tear volume. These animal studies showed that intralacrimal gland BoNT (both type A and B) injection leads to decreased tear production without prominent changes to the corneas and lacrimal glands.

## 5. Aqueous Deficiency: BoNT Reduces Lacrimal Gland Secretion in Humans

Although the lacrimal gland is thought to be innervated by sympathetic and parasympathetic nerves, the cholinergic parasympathetic system plays a more important role in tear production, including basal and reflex tear secretion [14]. Theoretically, BoNT blocks the release of acetylcholine from parasympathetic nerve endings in the lacrimal gland and subsequently decreases tear production.

In 1998, Boroojerdi et al. [4] first reported that BoNT could not only treat facial dyskinesia but also hyperlacrimation (i.e., “Crocodile tears”) in patients with aberrant regeneration of the facial nerve after facial palsy. Two patients with hyperlacrimation received a BoNT (20 mouse units) injection into the lacrimal gland and had a near complete recovery of tearing. This pioneering study investigated the effect of intraglandular BoNT injection on hyperlacrimation. Subsequently, several case reports showed satisfactory results for BoNT treatment of hyperlacrimation with minimal and transitory adverse events [15,16,17,18,19,20,21]. In these studies, 1–5 U of Botox^®^ (Allergan, Inc., Irvine, CA, USA) or 20 U of Dysport^®^ (Ipsen Ltd, Berks, UK) were injected into the lacrimal glands. BoNT effect duration ranged from three to six months and, in some cases, longer than one year [20]. In 2003, Whittaker et al. [22] used BoNT (2.5 to 5 U) to treat patients with functional epiphora and found that 8/11 (72.7%) patients had subjective and objective improvement in epiphora severity. In response to Whittaker et al. [22], some studies performed BoNT treatment (1.25 to 7.5 U) in patients with lacrimal outflow obstruction, including canalicular obstruction, punctal occlusion, and nasolacrimal duct obstruction, and reported an overall satisfaction rate of 70% [23,24,25,26,27,28]. Compared to conjunctivodacryocystorhinostomy (CDCR), a complete bypass of the lacrimal drainage system from inferior half of the caruncle to the middle nasal meatus, intraglandular BoNT (4 U) injection had superior accessibility, safety, reversibility, and technical simplicity [27]. Thus, BoNT injection into the lacrimal gland palpebral lobe is an alternative treatment choice for patients with lacrimal drainage obstruction who are not suitable for surgical intervention, such as older patients and those with multiple underlying diseases or malignancy of the lacrimal drainage system.

In recent years, several studies have reported decreased Schirmer test results following BoNT treatment of the lateral canthal rhytides [29,30,31]. Although Arat et al. [30] reported that Schirmer test results showed no significant differences after BoNT (10 U) injection into the crow’s feet area, 5/26 eyes showed a significant decrease in Schirmer test results from baseline. These findings demonstrated that injections into the lateral parts of the orbicularis oculi muscles can lead to reduced tear production and suggest that BoNT could diffuse from the orbicularis muscles and orbital septa into the lacrimal glands. To avoid BoNT effects on tear production during this aesthetic procedure, the lateral injection site should be at least 1 cm above and lateral to the orbital rim [32]. Moreover, BoNT diffusion depends on its concentration at the injection site. The diffusivity of BoNT increases at lower concentrations [33]. Thus, clinicians should use BoNT at higher concentrations (i.e., 5 to 10 U/0.1 mL) to limit the extent of the affected area. 

The aforementioned studies suggest that direct BoNT injection into the lacrimal gland and lateral parts of the orbicularis oculi muscle can impair aqueous tear production by the lacrimal gland. This effect may last three to six months on average with 1 to 5 U of Botox^®^ or 20 U of Dysport^®^.

## 6. Increased Tear Film Evaporation: Eyelid Malposition and Abnormal Blink Pattern

BoNT is beneficial for reducing involuntary eyelid contraction and blink rate in patients with facial movement disorders [34,35]. However, its use has several potential complications due to its chemodenervation effects on the orbicularis oculi muscles, such as lagophthalmos, incomplete blinks, and ectropion, which may lead to increased tear evaporation [36]. In the 1980s studies, lagophthalmos occurred in 63.6% of patients with blepharospasm who received BoNT treatment [37,38] and this complication rate decreased to 11.8–34.3% in recent reports [39,40]. However, this result might be an underestimate since lagophthalmos is generally measured in the upright position in which the effect of gravity on the upper eyelid is maximal. Moreover, our previous study showed that, at one month after BoNT injection, the rate of incomplete blinks was 78.8% in patients with facial movement disorders [39], which was higher than that of the normal population (i.e., 0.9–56.5%) [41].

Blinks are a protective mechanism for the ocular surface, which maintain corneal epithelial health and optical quality via the tear film. The apposition of the eyelids during a complete blink promotes lipid secretion from the meibomian glands. As a result, complete blinks not only distribute lipids across the tear film but also allow lipid expression from the meibomian glands [42]. Moreover, lipids help to stabilize the tear film and coat its aqueous part preventing evaporation. Therefore, incomplete blinking and lagophthalmos due to orbicularis oculi muscle paralysis contribute to unstable tear film and result in dry eye symptoms. Furthermore, incomplete blinking may lead to meibomian gland dysfunction (e.g., meibomian gland dropout or abnormal expressed meibum), which has been observed in patients with diseases affecting blinking (e.g., facial nerve palsy or thyroid-associated orbitopathy) and normal subjects [43,44,45,46]. Wang et al. [46] reported that 77 normal subjects with detectable incomplete blinks showed more dryness (P = 0.01), less tear film break-up time (TBUT; P = 0.04), and a higher grade of meibomian gland dropout (P < 0.001) compared to those without incomplete blinks. Although these complications are mild and transient, applying sufficient topical lubricants with or without an eye patch during sleep as well as frequently producing voluntary complete blinks during the day are recommended to avoid dry eye symptoms [47].

Ectropion, which has an overall incidence of <1%, is a less common complication of periocular BoNT injection that may cause EDE. Ectropion mechanisms include preceding horizontal laxity of lower eyelid structures, especially the lateral canthal tendon and subsequent paralysis of the orbicularis oculi muscles [36]. Ectropion can be avoided by being aware of pre-existing lower lid laxity prior to administering BoNT injections. In summary, before performing the BoNT injection, the identification of high-risk patients who are predisposed to eyelid malposition is necessary. When eyelid malposition and incomplete blinks occur, patients must be educated about supportive care, including topical lubricants and the performance of frequent complete blinking.

## 7. Dry Eye Improvement: BoNT Decreases Lacrimal Drainage

Despite BoNT’s well-known suppressive effect on lacrimal gland tear production in vivo and in vitro, a few studies in the late 1980s and early 1990s have reported epiphora after BoNT injection in patients with facial movement disorders [48,49,50]. According to these studies, epiphora occurred in 3.5% of treatments (range, 0–20%), which was less than the incidence of dry eye (range, 7.5–70%). In 1997, Spiera et al. [51] first described increased tear measurements assessed by the Schirmer test after treating blepharospasm with BoNT in patients with Sjögren’s syndrome. However, the authors could not identify a mechanism associated with their results at that time.

In 2000, Sahlin et al. [52] reported reduced mean blink outputs in patients with dry eye following BoNT-A (Botox^®^) injections. The injection of 3.75 U of Botox^®^ into the medial part of the lower eyelids resulted in a mean blink output reduction of up to 30% from baseline values (P < 0.001), whereas the injection of 2.5 U of Botox^®^ into the medial parts of the upper and lower eyelids resulted in a 62% reduction in the mean blink output (P < 0.001). These findings indicated that the injection of BoNT-A into the medial portion of the eyelids could lessen the efficacy of lacrimal drainage output by paralyzing the orbicularis oculi muscles directly acting on the canaliculi. Moreover, the authors suggested that decreased horizontal sliding of the lower eyelid and diminished vertical movement of the upper eyelid, which resulted in poor apposition of the puncta during blinks but also accounted for their results.

Subsequently, a few researchers have studied the efficacy and safety of BoNT-A injection into the medial eyelids [53,54,55,56,57,58]. Sahlin et al. [53] found the injection of BoNT-A into the medial lower eyelid reduced lacrimal pump and decreased ocular discomfort in a patient with Sjogren’s syndrome. Park and colleagues [54] investigated changes in tear production, distribution, and drainage after BoNT-A injection in 23 patients with blepharospasm. The authors found that dry eyes were ameliorated after treatment due to increased tear meniscus height, TBUT, and Tc-99m pertechnetate 50% clearance time caused by decreased muscle spasms and increased tear storage at the ocular surface. Moreover, Yang et al. [55] found that patients receiving BoNT (2.5 U of Botox^®^) injections into the medial lower eyelid had better TBUT and Schirmer test results than those who did not receive medial lower eyelid injections.

Bukhari et al. [56] compared the efficacy and side effects of BoNT and punctal plugs in 60 patients with dry eye. Compared to the punctal plug group, patients receiving 3.3 U of Botox^®^ injected into the medial lower eyelids had higher overall satisfaction (100%) and a lower complication rate (16.7%). Following this study, Fouda et al. [57] demonstrated similar findings for the treatment of patients with dry eye after laser in situ keratomileusis (LASIK) surgery. Sixty patients were randomly and equally assigned to one of three groups comprising the control (i.e., topical lubricants only), BoNT, and punctal plug (i.e., temporary silicone plugs in the lower puncta) groups. In the BoNT group, 3.75 U of Botox^®^ was injected into the medial lower eyelids to induce punctal ectropion immediately following LASIK surgery. Six weeks after LASIK surgery, the BoNT group had a significant increase in TBUT and Schirmer test results compared to the control group (TBUT: 7.01 ± 1.61 vs. 4.94 ± 1.23 s; Schirmer test: 8.4 ± 4.37 vs. 5.14 ± 1.91 mm). Moreover, the BoNT group had a lower complication rate than the punctal plug group (i.e., 25 vs. 60%).

Recently, a prospective, randomized, and a comparative eye-to-eye interventional study was conducted by Serna-Ojeda and colleagues [58] to evaluate the effect of BoNT (i.e., 4 U of Botox^®^ injected into the medial lower eyelid) on TBUT, Schirmer test results, and ocular surface staining in patients with dry eye. Unsurprisingly, the BoNT group had no complications and better results than the sham group. Moreover, these group differences maintained statistical significance at three months after treatment and then declined to no significance at six months.

In summary, the injection of BoNT (2.5–4 U of Botox^®^) into the medial portion of the eyelids, especially into the lower eyelids, can lead to temporary (i.e., < six months) paralysis of the pericanalicular orbicularis oculi muscle and impaired apposition of the puncta during blinking, resulting in poor lacrimal drainage system pumping force and subsequent tear retention. Thus, BoNT injection into the medial portion of the eyelids could be a treatment option to reduce tear drainage in patients with dry eye.

## 8. BoNT Effects on Tear Film Stability and Meibomian Gland Changes 

Since the early 1990s, dozens of studies have investigated the effect of BoNT on the lacrimal glands and tear production. However, only a few papers have focused on its effects on tear film stability and lipid tear deficiency following a 2007 report from the DEWS addressing the importance of meibomian gland dysfunction in DED [59]. In humans, the meibomian glands are thought to be regulated by the cholinergic parasympathetic nervous system [60]. Theoretically, BoNT blocks the release of acetylcholine from parasympathetic nerve endings and then impedes meibum production. Moreover, the meibomian glands are surrounded by pre-tarsal orbicularis oculi and Riolan’s muscles, which compress the acini and central ducts to spread meibum from the orifice into the tear film [42,61]. BoNT paralyzes these muscles and eventually decreases meibum excretion onto the ocular surface. Thus, BoNT should cause secretory and excretory meibum reduction and consequently decrease lipid layer thickness (LLT) and tear film stability. This hypothesis was supported by Ho et al. [31] who injected BoNT in patients with lateral canthal rhytides and showed significantly decreased tear film stability as assessed by TBUT at 1- and 3-months post-BoNT injection. However, in several studies, TBUT significantly increased following BoNT injection in patients with facial movement disorders [54,62,63,64,65]. Moreover, our previous data have shown a significant increase in LLT from baseline at 1-month post-treatment (89.9 ± 16.7 [one month] vs. 75.1 ± 21.2 [baseline], P = 0.01) [39]. These conflicting results may arise from decreased lacrimal drainage after BoNT injection into the medial portion of the eyelids, which impairs lacrimal drainage system pumping force and punctal apposition.

Regarding changes to the meibomian gland following BoNT injection, only one study has described how meibomian gland dropout did not change in patients with facial movement disorders at 1-month post-injection; however, the follow-up period may have been too short to reveal possible changes in meibomian gland morphology [39]. Further in vivo and in vitro studies are needed to examine the long-term effects of BoNT on the meibomian glands.

Overall, changes to the tear film lipid layer following BoNT injection are the same as those to the aqueous tear produced by the lacrimal gland. LLT and tear film stability may decrease with aesthetic usage and increase when BoNT is injected into the medial eyelid due to tear retention. However, long-term studies are needed to understand the effect of BoNT on the meibomian glands.

## 9. BoNT Influences the Corneal Parameter: Reduced Astigmatism and High-order Aberrations

Many factors contribute to the development of corneal astigmatism, and eyelid pressure and tension are two of the most important factors [66,67]. If the cornea is not affected by eyelid pressure, it shows against-the-rule (ATR) astigmatism (i.e., the horizontal corneal meridian is steepest), whereas with-the-rule (WTR) astigmatism (i.e., the vertical corneal meridian is steepest) occurs with high eyelid pressure. Thus, patients with blepharospasm and hemifacial spasm should have WTR astigmatism, and BoNT injection would diminish eyelid tension and further reduce WTR astigmatism. Moon et al. [66] found that WTR astigmatism was more common than ATR astigmatism in patients with blepharospasm and hemifacial spasm. A reduction in WTR astigmatism and an increase in ATR astigmatism were observed at one month after BoNT treatment; however, astigmatism levels returned to baseline at six months after injection. Moreover, Gunes et al. [63] found an ATR shift and decreased mean corneal astigmatism at three weeks and three months after BoNT treatment in patients with blepharospasm and hemifacial spasm.

Isshiki et al. [65] analyzed high-order aberrations (HOAs) in patients with blepharospasm using a wavefront aberrometer. These authors observed increased total HOA values and four HOA patterns, comprising stable (60.5%), small fluctuation (14.5%), sawtooth (17.1%), and reverse sawtooth (7.9%) patterns. Moreover, BoNT injection increased the stable HOA pattern to 94.7% and reduced the HOA values of all of the other patterns. These authors suggested that their findings might be due to the improvement of tear film status resulting from impaired lacrimal pumping, a decreased blink rate, and reduced friction between the eyelid and ocular surface. 

In summary, BoNT treatment changes corneal astigmatism by relieving eyelid pressure on the ocular surface and improves HOAs by increasing lacrimal tear status in patients with facial movement disorders.

## 10. Histological and Morphological Changes of the Lacrimal Gland and Ocular Surface

Harris and colleagues [68] first evaluated the histological features of the orbicularis oculi following BoNT injection in patients with blepharospasm and found no inflammation or persistent histological changes in the muscle fibers. In 2003, Horwath-Winter et al. [69] treated 16 patients with blepharospasm and dry eye using BoNT and investigated its effect on tear function, ocular surface morphology, and dry eye symptoms. Although significant decreases in Schirmer test values accompanied by slightly increased rose bengal staining was noted after BoNT injection, the impression cytology collected from the superior and inferior bulbar conjunctiva was identical pre- and post-treatment.

Moreover, a few animal studies have investigated histological changes to the lacrimal gland and ocular surface after intralacrimal gland BoNT injection and had similar results [9,10,11,13]. Suwan-apichon et al. [9] induced a mouse dry eye model by injecting different amounts of BoNT-B (1.25, 5, or 20 mU; Myobloc^®^) into the lacrimal gland. Although significantly decreased tear production and increased corneal fluorescein staining were observed, histological analyses showed no lymphocytic or inflammatory cell infiltration into the conjunctiva or lacrimal gland stroma or acini. Hematoxylin and eosin (H&E) and immunofluorescent staining of the mouse lacrimal gland by Park et al. [11] also showed normal acinar structure without inflammatory cell or T-cell infiltration. Other studies found no inflammation or architectural changes in the rabbit lacrimal gland after BoNT-A injection [10,13]. Thus, clinical and animal studies have shown that intramuscular and intraglandular injections of BoNT did not cause histological changes, particularly inflammation, and were safe for the injection area.

## 11. Inflammatory Cytokine Changes

Several studies have analyzed inflammatory cytokines in BoNT-induced animal dry eye models and found increased gene expression levels of certain cytokines. Park et al. [11] injected BoNT-B into the murine lacrimal gland and found higher macrophage migration inhibitory factor (MIF) and IL-12 levels in the lacrimal gland. Despite elevated levels of MIF, a potent activator of T lymphocytes in many inflammatory diseases, the lacrimal gland maintained its architecture without prominent inflammatory cell infiltration. This result suggests that these inflammatory cytokines were secreted into the tear film rather than diffused into the lacrimal gland. Zhu et al. [12] analyzed cytokines from the conjunctival and corneal epithelium in BoNT-B-treated mice and found significantly elevated TNF-α and IL-1β expression at 1- and 2-wk post-BoNT injection compared to control groups. However, IL-1β levels returned to baseline at four weeks. This finding was also associated with decreased tear production and increased corneal fluorescein staining. Kim et al. [13] found decreased tear production as well as increased EGF expression and concentration in rabbits that underwent intralacrimal gland BoNT-A injection. EGF plays a key role in stimulating lacrimal gland secretion and modifying the neural regulation of lacrimal gland protein secretion [70,71]. From these results, the authors inferred that decreased tear volume after BoNT injection leads to the activation of ocular surface sensory nerves and stimulation of the parasympathetic and sympathetic nerves of the lacrimal gland. Thus, EGF synthesis was stimulated via a series of signaling cascades.

In 2014, Lu et al. [64] reported elevated inflammatory cytokines levels, including TNF-α, IL-1β, IL-6, IL-2, IL-17, and vascular endothelial growth factor, in the tear film of patients with blepharospasm and DED. Of these cytokines, the levels of IL-6 and IL-17 significantly decreased after periocular injection of 17.5 U of BoNT-A (Botox^®^). These authors suggested that BoNT not only relieved patients muscle spasms but also improved their ocular surface condition due to increases in tear film stability and decreases in inflammatory cytokine levels.

There was an inconsistency between animal and clinical studies. In the animal models, we assumed that BoNT was injected into normal lacrimal glands, which led to elevated inflammatory cytokine levels that were compared to normal groups. However, in the clinical study by Lu and colleagues [64], inflammatory cytokine levels were compared before and after BoNT injection in patients with blepharospasm and DED only. Therefore, no post-BoNT injection data could be obtained from a normal control group. In summary, BoNT may change certain inflammatory cytokine levels in the lacrimal gland and ocular surface. However, this observation is not associated with structural changes or actual inflammation in the injected tissues.

## 12. Conclusions

BoNT has been widely used in ophthalmology and the aesthetic industry, where it has altered clinical practice concerning the management of facial movement disorders, autonomic disorders, and aesthetic wrinkles. Several side effects have been reported after periocular BoNT injection. However, the effect of BoNT on the tear film homeostasis remained controversial to date. This review concludes that tear film homeostasis after periocular BoNT injection depends on dosage, concentration, and the most important factor—injection site of BoNT. Therefore, clinicians should adopt different treatment protocols for each patient according to his/her ocular surface condition to prevent potential complications or avoid exacerbating pre-existing comorbidities. This requires familiarity with BoNT and thorough ocular examination of each patient. If it is properly administered, BoNT can be an effective and safe tool.

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
