# Peer review of "A Review of Periocular Botulinum Neurotoxin on the Tear Film Homeostasis and the Ocular Surface Change"

_toxins, 2019, doi:10.3390/toxins11020066_

Round 1
Reviewer 1 Report
I read the review MS entitled "A review of periocular botulinum neurotoxin on the tear film homeostasis and the ocular surface change" which gives an extensive analysis of the recent and not recent literature about the effect of Botulin toxin periocular injection on tear film homeostasis.
The topic, is very interesting, because it analyses both the collateral effects of Botulin toxin on tear homeostasis by analyzing different dysfunctional causative effects, and the use of Botulin toxin to solve some diseases primarily due to tear homeostasis dysfunction.
I found the paper well written and clearly organized. It is well suited to the editorial policy of this Journal open not only to basic science but also to clinical studies and clinical therapy.
For these reasons I suggest its publication.
Author Response
We thank the reviewer for spending time reviewing this manuscript and highly appreciate his/her consideration.
Reviewer 2 Report
Review of toxins-428763-peer-review-v1, 14th January 2019
This is a well written review. The following comments should be used to improve the final version.
Note to the authors: The use of serotype B BoNT might yield completely different results to that of serotype A due to the different SNARE target. Also required doses of type B for humans (as opposed to animals) are known to be very significantly higher than type A due to receptor differences. Therefore, there should be caution in interpreting results obtained for the two serotypes in the same way.
Abstract
This is too long and should be reduced in length.
Introduction
Line 29 The term “cosmetic” should not be used as BoNT is not a cosmetic product.
Reference 5 is very old and should be replaced by a more current version, of which there are many.
Lines 32-33 BoNT has several different and distinct activities which have been elucidated over the years and so the authors should cite the BoNT activity described as an example only.
Line 43 The authors should state when they performed the searches as an important reference on tear production and BoNT has not been included, presumably due to publication after their searches were complete (Alsuhaibani and Eid 2018).
Subsequent sections
Line 51 A reference is needed to this report.
Line 54 DED should be written in full at this first citation.
Section 4 The authors have not referred to the studies in their references 9-12. In fact, these references have not been cited anywhere in the manuscript. This should be corrected.
Line 78 BoNT does not inhibit, it blocks (as stated in line 32).
Line 95 It would help the reader if a brief description of CDCR was included.
Line 108 The authors use the term “spreading” here but “diffusion” in line 105. The correct term is diffusion. Spreading is a physical injection phenomenon (Pickett 2009).
Section 5 It would be useful if the authors indicated that there have been no dose-ranging studies performed in adults and therefore doses they highlight in line 113 are only arbitrary.
Line 115-6 A reference is needed to this statement.
References 35 & 36 date back to 1985. The authors should state this and that the effects seen in these publications probably reflect the earliest usages of BoNT rather than more modern results.
Line 126 Is “apposition” the correct term to use here?
Reference 39 Perhaps a more up to date reference should be used here.
Line 139 A reference is needed for these recommendations.
Line 140 et seq References are needed in this section.
References 44-46 Please see comment above on references 35 & 36.
Line 157 The BoNT product should be named here. Please could the authors check that, wherever they have named units of BoNT, the product name is given.
Line 165 et seq The authors have discussed references 50-54 in detail but have omitted reference 49. Why?
Line 168 TBUT should be written in full here.
Line 184 et seq The sham group should be described here.
Line 197-8 This statement should be more supported – number of studies performed? Countries? Conditions studied?
Line 199 Is this the same reported cited in line 51?
Line 201 Same comment as previously to line 78.
Lines 204-5 Is there any evidence to support this claim of muscle paralysis or is this speculation by the authors?
Lines 226-7 Reference needed here.
Lines 227-9 The authors should include a little explanation of these terms to help the reader.
Lines 247, 268 & 301 The section numbering is incorrect here.
Lines 248- 255 More recent citations are needed in this section
Line 266 “relatively safe” as measured by……?
Lines 281-3 A reference is needed here.
Conclusion
The authors have not concluded anything about the title subject of their paper. This should be changed.
Alsuhaibani, A. H. and S. A. Eid (2018). "Botulinum toxin injection and tear production." Curr Opin Ophthalmol 29(5): 428-433.
Pickett, A. (2009). "Dysport: pharmacological properties and factors that influence toxin action." Toxicon 54(5): 683-689.
Author Response
Note to the authors: The use of serotype B BoNT might yield completely different results to that of serotype A due to the different SNARE target. Also required doses of type B for humans (as opposed to animals) are known to be very significantly higher than type A due to receptor differences. Therefore, there should be caution in interpreting results obtained for the two serotypes in the same way.
Response to note: We thank the reviewer’s reminder. We’ve amended the summary of Section 4 as following.
These animal studies showed that intralacrimal gland BoNT (both type A and type B) injection leads to decreased tear production without prominent changes to the corneas and lacrimal glands (Page 2, section 4).
Point 1: Abstract: This is too long and should be reduced in length.
Response 1: We thank the reviewer for this recommendation. We’ve modified Abstract a little bit (Word counts: 198 to 154).
Point 2: Line 29 The term “cosmetic” should not be used as BoNT is not a cosmetic product.
Response 2: We thank the reviewer for the recommendation. We’ve changed “cosmetic” to “facial wrinkles” in order not to confuse the readers.
Point 3: Reference 5 is very old and should be replaced by a more current version, of which there are many.
Response 3: We thank the reviewer for this suggestion. A new Reference 5 is as following.
1. Carruthers, J., et al., Consensus recommendations on the use of botulinum toxin type a in facial aesthetics. Plast Reconstr Surg, 2004. 114(6 Suppl): p. 1S-22S.
Point 4: Lines 32-33 BoNT has several different and distinct activities which have been elucidated over the years and so the authors should cite the BoNT activity described as an example only.
Response 4: We thank the reviewer for his/her circumspection and are sorry for our negligence. We’ve already cited two references about BoNT activity.
1. Jankovic, J., Botulinum toxin in clinical practice. J Neurol Neurosurg Psychiatry, 2004. 75(7): p. 951-7.
2. Humeau, Y., et al., How botulinum and tetanus neurotoxins block neurotransmitter release. Biochimie, 2000. 82(5): p. 427-46.
Point 5: Line 43 The authors should state when they performed the searches as an important reference on tear production and BoNT has not been included, presumably due to publication after their searches were complete (Alsuhaibani and Eid 2018).
Response 5: We agree and thank the reviewer for this concern. We’ve already added the date on PubMed search (14th July 2018) to the manuscript.
Point 6: Line 51 A reference is needed to this report.
Response 6: We thank the reviewer for his/her circumspection. We’ve already cited the reference about TFOS DEWS II report.
1. Craig, J.P., et al., TFOS DEWS II Definition and Classification Report. Ocul Surf, 2017. 15(3): p. 276-283.
Point 7: Line 54 DED should be written in full at this first citation.
Response 7: We thank the reviewer for this suggestion. We’ve already corrected this mistake.
Point 8: Section 4 The authors have not referred to the studies in their references 9-12. In fact, these references have not been cited anywhere in the manuscript. This should be corrected.
Response 8: We thank the reviewer for his/her circumspection. We’ve added the description of reference 9 and 11 in this section and deleted reference 10 and 12.
Point 9: Line 78 BoNT does not inhibit, it blocks (as stated in line 32).
Response 9: We thank the reviewer for this suggestion and have amended the sentence.
Point 10: Line 95 It would help the reader if a brief description of CDCR was included.
Response 10: We thank the reviewer for the recommendation. A description of CDCR has been added into the manuscript (Page 3, section 5).
Point 11: Line 108 The authors use the term “spreading” here but “diffusion” in line 105. The correct term is diffusion. Spreading is a physical injection phenomenon (Pickett 2009).
Response 11: We thank the reviewer for the suggestion and have corrected this term in the manuscript.
Point 12: Section 5 It would be useful if the authors indicated that there have been no dose-ranging studies performed in adults and therefore doses they highlight in line 113 are only arbitrary.
Response 12: We thank the reviewer for this concern. Although there have been no dose-ranging studies performed in human, there were several case series (as following) reported the duration of BoNT on hyperlacrimation. In these studies, 1-5 U of Botox® or 20 U of Dysport® were injected into the lacrimal glands. BoNT effect duration ranged from three to six months. Therefore, we summarized these results in line 113 (Page 3, Section 5, paragraph 4).
1. Riemann, R., et al., Successful treatment of crocodile tears by injection of botulinum toxin into the lacrimal gland: a case report. Ophthalmology, 1999. 106(12): p. 2322-4.
2. Hofmann, R.J., Treatment of Frey's syndrome (gustatory sweating) and 'crocodile tears' (gustatory epiphora) with purified botulinum toxin. Ophthalmic Plast Reconstr Surg, 2000. 16(4): p. 289-91.
3. Keegan, D.J., et al., Botulinum toxin treatment for hyperlacrimation secondary to aberrant regenerated seventh nerve palsy or salivary gland transplantation. Br J Ophthalmol, 2002. 86(1): p. 43-6.
4. Montoya, F.J., et al., Treatment of gustatory hyperlacrimation (crocodile tears) with injection of botulinum toxin into the lacrimal gland. Eye (Lond), 2002. 16(6): p. 705-9.
5. Nava-Castaneda, A., et al., Duration of botulinum toxin effect in the treatment of crocodile tears. Ophthalmic Plast Reconstr Surg, 2006. 22(6): p. 453-6.
6. Ito, H., et al., Low-dose subcutaneous injection of botulinum toxin type A for facial synkinesis and hyperlacrimation. Acta Neurol Scand, 2007. 115(4): p. 271-4.
7. Girard, B., et al., Botulinum neurotoxin A injection for the treatment of epiphora with patent lacrymal ducts. J Fr Ophtalmol, 2018. 41(4): p. 343-349.
Point 13: Line 115-6 A reference is needed to this statement.
Response 13: We thank the reviewer for his/her circumspection. We’ve already cited the reference.
1. Kenney, C. and J. Jankovic, Botulinum toxin in the treatment of blepharospasm and hemifacial spasm. J Neural Transm (Vienna), 2008. 115(4): p. 585-91.
2. Ferrazzano, G., et al., Botulinum toxin and blink rate in patients with blepharospasm and increased blinking. J Neurol Neurosurg Psychiatry, 2015. 86(3): p. 336-40.
Point 14: References 35 & 36 date back to 1985. The authors should state this and that the effects seen in these publications probably reflect the earliest usages of BoNT rather than more modern results.
Response 14: We thank the reviewer for this recommendation. We’ve added a more modern result in this paragraph.
1. Ababneh, O.H., A. Cetinkaya, and D.R. Kulwin, Long-term efficacy and safety of botulinum toxin A injections to treat blepharospasm and hemifacial spasm. Clin Exp Ophthalmol, 2014. 42(3): p. 254-61.
Point 15: Line 126 Is “apposition” the correct term to use here?
Response 15: Yes, this term was used by Sahlin et al., and the original sentence was “It is also presumed that the apposition of the puncta during blinking is an important part of the valve mechanism in the lacrimal pump.”
1. Sahlin, S., et al., Effect of eyelid botulinum toxin injection on lacrimal drainage. Am J Ophthalmol, 2000. 129(4): p. 481-6.
Point 16: Reference 39 Perhaps a more up to date reference should be used here.
Response 16: We thank the reviewer for the recommendation and changed the reference to a more recent one.
1. Knop, E., et al., The international workshop on meibomian gland dysfunction: report of the subcommittee on anatomy, physiology, and pathophysiology of the meibomian gland. Invest Ophthalmol Vis Sci, 2011. 52(4): p. 1938-78.
Point 17: Line 139 A reference is needed for these recommendations.
Response 17: We thank the reviewer for his/her circumspection. We’ve already cited the reference.
1. McMonnies, C.W., Incomplete blinking: exposure keratopathy, lid wiper epitheliopathy, dry eye, refractive surgery, and dry contact lenses. Cont Lens Anterior Eye, 2007. 30(1): p. 37-51.
Point 18: Line 140 et seq References are needed in this section.
Response 18: We thank the reviewer for his/her reminder. The reference (Dutton et al.) should be cited at the end of the second sentence in this section instead of the first one.
1. Dutton, J.J. and A.M. Fowler, Botulinum toxin in ophthalmology. Surv Ophthalmol, 2007. 52(1): p. 13-31.
Point 19: References 44-46 Please see comment above on references 35 & 36.
Response 19: We thank the reviewer for this recommendation. We’ve added a description of “in the late 1980s and early 1990s” in this section.
Point 20: Line 157 The BoNT product should be named here. Please could the authors check that, wherever they have named units of BoNT, the product name is given.
Response 20: We thank the reviewer for this suggestion. The BoNT product name was Botox® and we replaced ‘BoNT’ with ‘Botox®’ in this paragraph.
Point 21: Line 165 et seq The authors have discussed references 50-54 in detail but have omitted reference 49. Why?
Response 21: We thank the reviewer for pointing out our negligence. We’ve added the description of reference 49 as following.
Sahlin et al. [53] found the injection of BoNT-A into the medial lower eyelid reduced lacrimal pump and decreased ocular discomfort in patient with Sjogren’s syndrome.
Point 22: Line 168 TBUT should be written in full here.
Response 22: We thank the reviewer for this suggestion. The full name of TBUT is tear break-up time, which is first described in the section 6, paragraph 2 (Wang et al. [46] reported that 77 normal subjects with detectable incomplete blinks showed more dryness (P = 0.01), less tear film break-up time (TBUT; P = 0.04)).
Point 23: Line 184 et seq The sham group should be described here.
Response 23: We thank the reviewer for this recommendation. We added the data of TBUT and Schirmer test in BoNT group and control group (TBUT: 7.01 ± 1.61 vs 4.94 ± 1.23 s; Schirmer test: 8.4 ± 4.37 vs 5.14 ± 1.91 mm).
Point 24: Line 197-8 This statement should be more supported – number of studies performed? Countries? Conditions studied?
Response 24: We thank the reviewer for this suggestion. More than 20 studies (including animal and human studies) investigated BoNT effect on tear production of the lacrimal gland, but most of these studies didn’t investigate tear film stability. This statement is a short description of the sections mentioned previously (i.e. reduced tear production) and we want to emphasize the importance of tear film stability and meibomian gland function in dry eye disease.
Point 25: Line 199 Is this the same reported cited in line 51?
Response 25: No, it’s not. This report published in 2007 and was an earlier report of the International Dry Eye Workshop.
1. The definition and classification of dry eye disease: report of the Definition and Classification Subcommittee of the International Dry Eye WorkShop (2007). Ocul Surf, 2007. 5(2): p. 75-92.
Point 26: Line 201 Same comment as previously to line 78.
Response 26: We thank the reviewer for this suggestion and have amended the sentence.
Point 27: Lines 204-5 Is there any evidence to support this claim of muscle paralysis or is this speculation by the authors?
Response 27: Yes. Riolan’s muscle is a subdivision of striated muscle which is separate from the pretarsal orbicularis oculi muscle. Therefore, BoNT could paralyze these muscles and further impair meibum excretion. Two references were added into this section.
1. Lipham, W.J., H.A. Tawfik, and J.J. Dutton, A histologic analysis and three-dimensional reconstruction of the muscle of Riolan. Ophthalmic Plast Reconstr Surg, 2002. 18(2): p. 93-8.
2. Knop, E., et al., The international workshop on meibomian gland dysfunction: report of the subcommittee on anatomy, physiology, and pathophysiology of the meibomian gland. Invest Ophthalmol Vis Sci, 2011. 52(4): p. 1938-78.
Point 28: Lines 226-7 Reference needed here.
Response 28: We thank the reviewer for his/her reminder. Two references were cited here.
1. Moon, N.J., H.I. Lee, and J.C. Kim, The changes in corneal astigmatism after botulinum toxin-a injection in patients with blepharospasm. J Korean Med Sci, 2006. 21(1): p. 131-5.
2. Zinkernagel, M.S., A. Ebneter, and D. Ammann-Rauch, Effect of upper eyelid surgery on corneal topography. Arch Ophthalmol, 2007. 125(12): p. 1610-2.
Point 29: Lines 227-9 The authors should include a little explanation of these terms to help the reader.
Response 29: We thank the reviewer for his/her friendly reminder. We’ve added a description of against-the-rule (i.e. the horizontal corneal meridian is steepest) and with-the-rule (i.e. the vertical corneal meridian is steepest) astigmatism in the manuscript.
Point 30: Lines 247, 268 & 301 The section numbering is incorrect here.
Response 30: We thank for the reviewer for pointing out the mistake. We renumbered these 3 sections in the manuscript.
Point 31: Lines 248- 255 More recent citations are needed in this section.
Response 31: We agree with the reviewer entirely. However, Harris et al. and Horwath-Winter et al. are the only 2 studies investigating histological changes of the lacrimal gland and ocular surface of human.
Point 32: Line 266 “relatively safe” as measured by……?
Response 32: From the results of both human and animal studies, injection of BoNT didn’t cause localized morphological changes or inflammation of the ocular surface. So BoNT was safe for the injection area. In order not to confuse the readers, we modified the sentence from “injections of BoNT were relatively safe and did not cause histological changes, particularly inflammation, in the injection area” to “injections of BoNT did not cause histological changes, particularly inflammation, and were safe for the injection area”.
Point 33: Lines 281-3 A reference is needed here.
Response 33: We thank the reviewer for his/her reminder. Two references were cited here.
1. Wilson, S.E., S.A. Lloyd, and R.H. Kennedy, Epidermal growth factor messenger RNA production in human lacrimal gland. Cornea, 1991. 10(6): p. 519-24.
2. Tepavcevic, V., et al., Signal transduction pathways used by EGF to stimulate protein secretion in rat lacrimal gland. Invest Ophthalmol Vis Sci, 2003. 44(3): p. 1075-81.
Point 34: Conclusion: The authors have not concluded anything about the title subject of their paper. This should be changed.
Response 34: We thank the reviewer for this recommendation. We’ve modified Conclusion to meet the title of this manuscript.
